# Massive Data Storage Solution for IoT Devices Using Blockchain Technologies

**DOI:** 10.3390/s23031570

**Published:** 2023-02-01

**Authors:** Alexandru A. Maftei, Alexandru Lavric, Adrian I. Petrariu, Valentin Popa

**Affiliations:** Computers, Electronics and Automation Department, Stefan cel Mare University of Suceava, 720229 Suceava, Romania

**Keywords:** blockchain, consensus mechanisms, secure data management, data storage, Internet of Things, wireless sensor networks

## Abstract

The Internet of Things (IoT) concept involves connecting devices to the internet and forming a network of objects that can collect information from the environment without human intervention. Although the IoT concept offers some advantages, it also has some issues that are associated with cyber security risks, such as the lack of detection of malicious wireless sensor network (WSN) nodes, lack of fault tolerance, weak authorization, and authentication of nodes, and the insecure management of received data from IoT devices. Considering the cybersecurity issues of IoT devices, there is an urgent need of finding new solutions that can increase the security level of WSNs. One issue that needs attention is the secure management and data storage for IoT devices. Most of the current solutions are based on systems that operate in a centralized manner, ecosystems that are easy to tamper with and provide no records regarding the traceability of the data collected from the sensors. In this paper, we propose an architecture based on blockchain technology for securing and managing data collected from IoT devices. By implementing blockchain technology, we provide a distributed data storage architecture, thus eliminating the need for a centralized network topology using blockchain advantages such as immutability, decentralization, distributivity, enhanced security, transparency, instant traceability, and increased efficiency through automation. From the obtained results, the proposed architecture ensures a high level of performance and can be used as a scalable, massive data storage solution for IoT devices using blockchain technologies. New WSN communication protocols can be easily enrolled in our data storage blockchain architecture without the need for retrofitting, as our system does not depend on any specific communication protocol and can be applied to any IoT application.

## 1. Introduction

The Internet of Things (IoT) concept involves connecting devices via the internet and forming a network of objects that can collect information from the environment without human intervention. According to [1], the number of IoT devices deployed from year to year is constantly increasing and it is estimated to reach 25.44 billion by 2030. There have been many surveys [2,3] conducted regarding the usage of different blockchain applications. Areas such as healthcare [4], agriculture [5], the smart grid [6], the smart city [7], or traffic management [8] are all contributing to the rapid growth of active IoT devices, as it seeks to automate the tasks of collecting data from the environment for further processing and storage.

The use of IoT devices in our daily life brings benefits such as the increase in automated tasks, obtaining low energy consumption and efficient resource management, automated data retrieval, information storage, and remote control of devices installed on the field. The IoT concept is usually implemented through wireless sensor networks (WSNs). Although the IoT concept offers some advantages, it also has some issues that are usually associated with cyber security risks, such as the lack of detection of malicious WSN nodes [9], lack of fault tolerance, weak authorization, and authentication of nodes [10], and the insecure management of retrieved data from IoT devices. Considering the cybersecurity issues for IoT devices, there is an urgent need of finding new solutions that can increase the security level of WSNs.

A WSN refers to a group of spatially dispersed wireless sensors distributed over a large geographical area that collect data from the environment and then send them to a sink node for processing and storage purposes. Using different visualization algorithms, the collected information is presented to users. The architecture of such a wireless sensor network can be seen in Figure 1. The structure includes a centralized sink node (Gateway) that is usually connected to the internet, and wireless sensors nodes, which is a communication mechanism that allows the information to be transmitted to the end users.

The communication protocol is specific to the application IoT type and usually is represented by Bluetooth [11], Wi-Fi [12], LoRaWAN [13], ZigBee [14], or NB-IoT [15] when we are considering low power consumption efficiency. Using multiple types of wireless communications protocols in the same wireless network can be both an advantage and a disadvantage. One advantage is the ability to use area-specific types of communications, such as Wi-Fi and Bluetooth for shorter-range communications, 10–50 m, or LoRaWAN and NB-IoT for long-range communications, 1–10 km, in urban environments [8]. One disadvantage that can arise from using multiple communication protocols in the same IoT network is interoperability [16]. This can become an issue because each communication protocol has its security policies for securing data, which can lead to congestion and slow down the entire network. Taking into account the large number of wireless communication protocols used in the IoT, it is somewhat difficult to maintain and update but also to detect problems that may occur in an IoT network due to the use of a large number of wireless protocols.

When we talk about WSNs, we consider networks where the number of active IoT devices can be hundreds of thousands or even millions, which will result in a large number of data packets that need to be stored securely. In a classic WSN network, all information is sent to a central data storage point. In terms of cybersecurity, centralized storage of large amounts of data is becoming a secure target for malicious entities. Another aspect of wireless sensor networks is that they have limited resources such as memory, computing power, and limited energy/battery capacity [17]. However, the role of WSN IoT devices is to operate for long periods, with as little human intervention as possible, being mostly standalone devices that are disposable at the end of their life.

The rest of the paper is organized as follows: Section 3 presents the blockchain technology along with its mechanisms that can provide security to data from IoT devices. Section 4 presents the proposed architecture for securing IoT data. Section 5 presents a proof-of-concept practical implementation of the proposed architecture, and the last section presents the conclusions.

The main contributions of this paper are the following:We present a solution for the secure management and storage of the data retrieved from IoT devices using blockchain technology.The data storage process is performed by using a dual blockchain topology that includes a lightweight blockchain (local blockchain) and a public blockchain. The lightweight blockchain stores temporarily all the IoT information acting as a buffer that stores the node identity ledgers and the hash addresses that point to the data packets located in the public blockchain acting as a register. The public blockchain permanently stores the entire IoT data stream sent through the entire WSN architecture.Our novel blockchain architecture uses an IoT authentication process that allows new WSN nodes to be accepted using a voting process that integrates the PoS consensus mechanism specifically used by blockchain architectures.Another advantage is related to the scalability of the proposed architecture, which can integrate a very large number of IoT devices without decreasing the performance level of the system. This IoT device can join the network and contribute to its maintenance by implementing the consensus mechanism.We propose an architecture where different WSN entities (e.g., gateways, sensors) have blockchain capabilities and functionalities to achieve a massive data storage solution.The proposed architecture is scalable and is not locked on a particular IoT communication protocol. New IoT sensors and communication protocols can be easily enrolled in our blockchain architecture without the need for retrofitting.Our massive blockchain data storage solution can also be used in a hybrid manner by classic IoT WSN networks with no enhanced capabilities.

## 2. Related Works

Blockchain technology can be used to implement different security aspects such as access control, authorization, and authentication processes of IoT devices in a wireless sensors network, methods to detect malicious devices or DDOS attacks in a network or achieve secure storage of IoT data. There are a few solutions in the literature that attempt to solve each problem individually. Thus, in this section, we present some solutions that aim to solve the problem of secure storage of data from IoT devices, solutions based on blockchain technology with all its features such as consensus mechanisms, smart contracts, decentralization, pseudo-anonymity (even anonymity in some cases), and immutability.

In [18], Liu et al. propose a Data Integrity as a Service (DIaaS) framework based on blockchain technology for verifying data coming from IoT devices. The authors attempt to solve two problems: the first is to eliminate the requirement of trust in third-party auditors (TPA) and increase the reliability of the data integration service. The second problem is that of verifying data without relying on a single third-party auditor, so protocols are proposed to verify data integrity in a fully decentralized environment. The main disadvantage of this approach is the fact that the integrity of the data packets needs to be verified by a trusted third-party authority.

In [19], Li et al. propose a scheme using blockchain technology for storing and protecting large amounts of data from IoT devices. Their proposed method guarantees data protection by having a large number of blockchain miners handle data from IoT devices, eliminating centralized servers. The proposed architecture uses edge computing to take over the task of processing the data and later sending it to the Distributed Hash Tables (DHT). A final proposal of the authors in their scheme is to use certificate-less cryptography. Certificateless cryptography reduces the redundancy that is brought about by traditional Public Key Infrastructure (PKI) and provides an efficient way of authentication for IoT devices. However, this scheme may have problems if there is a necessity for implementing a system that uses more complicated access control policies.

In [20], Shafagh et al. propose a system based on blockchain technology that provides access control and data management in a distributed manner. Among the contributions presented in the paper, we can mention designing a secure cryptographic method of sharing data with frequent key updates, the ability to revoke access to data, an efficient search method in a compressed chunked data stream, and a location-aware level of data storage. One drawback of the proposed architecture is the consensus mechanism used, namely the Proof of Work mechanism. PoW is a resource-intensive mechanism and suffers from the so-called 51% attack [21] that can occur if the proposed architecture does not have many users and hash power is abundantly available.

In another paper [22], Ren et al. propose an architecture for securely storing data from edge devices. This architecture uses blockchain technology; specifically, the authors implement two blockchain networks, one local and the other global. The local blockchain network has limited storage space and is created by the main edge nodes and stores all data coming from IoT devices. The global blockchain network is built using cloud servers and stores all data coming from local blockchain networks. The cloud servers calculate hash values for the data uploaded to the global blockchain, and to ensure the integrity of the data, a periodic check is performed using the hash values already calculated.

Ren et al. [23] propose the use of blockchain technology for storing data from Wireless Body Area Networks (WBANs). A modification added to the blockchain-based storage system is to implement a sequential aggregate signature scheme (DVSSA). This scheme ensures that data can only be accessed by assigned individuals and that the privacy of users in a WBAN network is protected. DVSAA can also compress the data stored in the blockchain, thus solving the problem of limited storage space.

Our blockchain data storage system differs from the works mentioned above in that it can accept other WSNs and IoT devices that do not have blockchain capabilities but wish to use the proposed ecosystem for secure and traceable data storage. All verification processes of the proposed architecture are performed by the IoT devices that join the network and want to contribute to the better functioning of the network; therefore, the ecosystem can work without interruption and user intervention.

In addition, in our work, we propose the use of the “Proof of Stake” consensus mechanism, a mechanism that eliminates the need to use devices with a high computing power because the nodes that add transactions in blocks and the blocks in the blockchain become validators that stake a part of their reputation rather than needing high computational power. Once again, the system proposed in this paper is a scalable one where a large number of IoT devices can join the network, contribute to its maintenance by implementing the consensus mechanism, or use the network only to send and securely store sensor data. The security of the system is also high because the more nodes there are that put their reputation on the line to validate blocks and transactions, the more secure your network will become. The number of transactions is also high, and this is again due to the PoS consensus mechanism, which does not require a high computing power, and due to this, energy consumption also decreases, thus reducing the carbon footprint.

## 3. Blockchain Technology for IoT Concept

Blockchain technology [24] involves chronologically saving data in the form of a blockchain called a distributed ledger where each block of data is linked to each other using cryptographic methods, thus making the entire system secure, immutable, and tamper-proof. A distributed ledger can be seen as a database that saves data in chronological order, but the difference is that of the permissions set on the data stored in the ledger. The main purpose of this paper is to propose an architecture of data storage using blockchain technology that can be a secure data management solution for the IoT concept.

Blockchain technology became known in 2008 when Satoshi Nakamoto presented it for the first time [25] as a way to introduce a virtual currency-sharing system through a Peer-to-Peer (P2P) network, thus eliminating centralized tertiary institutions. Over time, blockchain technology has evolved positively and has been increasingly adopted in various fields, with one of them being the Internet of Things. The implementation of the blockchain in different fields is due to the way the data are stored but also to the fact that this technology has several features that offer increased security. Each block in turn has three basic elements:

**Hash of the current block**. This is the result of applying a hash function to the data and information that is stored inside the block. A hash function is a one-way function that involves transforming a text of variable length into a text of constant size. The result of a hash function is also called a hash value or just a hash. Hash functions are used because they are computationally difficult to reverse. It should be noted that hash values are very sensitive to changes, that is, when a single character or number is changed, the whole hash value will become completely different.

**The encapsulated data in each block**. The data are passed through the hash function so that they cannot be read by anyone. Each block contains several data transactions that are measured in transactions per second (TPS), and this differs from system to system. In the case of the Bitcoin blockchain, which only saves information on the transfer of BTC virtual currencies, the number of transactions per second is approximately 7 TPS [26]. To add blocks of data, they must first go through a mining process, where users of a network compete to find the result of a math problem, in the case of the Bitcoin network. This mining process is due to the feature called the consensus mechanism. In the IoT field, the PoW mining process or PoS validation process can be performed by IoT devices. This operation of implementing consensus mechanisms can be applied by the IoT nodes of the network, thus providing the opportunity for IoT devices to manage the addition of data packets in blocks and then their addition to the blockchain.

**Hash of the previous block**. This is an important element for the security of the blockchain. Each block in the blockchain points to the hash of the previous block. If a block’s previous hash value is wrong, that block and its successors will become invalid. To modify a block, the hash value of the block must be recalculated together with other blocks, and this is performed using consensus mechanisms that can be difficult from a computational point of view.

Therefore, the way data are stored in the blockchain is a secure solution for saving data from wireless sensor networks. In addition, blockchain technology also has some features that can be beneficial for data storage, such as:*A*.Immutability

The blockchain is similar to a database but differs in the way the data are managed, and this is due to the immutability feature. Immutability refers to the fact that once data have been stored in a block, it cannot be modified in any way. Thus, the impossibility of modifying the data in a block introduces benefits such as traceability or audibility.

*B*.Decentralization and distributivity

In most cases, these two terms, decentralization and distributivity [27], are used interchangeably, although their meanings are different from each other. These two features are very important for the blockchain because any system that wants to implement this technology will automatically benefit from both features, so there is no need to implement other third-party technologies to act as data storage and management.

At present, most data storage and management applications use a server-client architecture similar to IoT classical data storage. In such an architecture, the transfer of data between two or more clients is performed through a server that acts as an intermediary for forwarding information. Thus, if the server suffers a cyber-attack, the entire system may be left without the central data distribution system. To solve this problem, the distributive feature of blockchain technology can be used, which uses a peer-to-peer (P2P) architecture where the transfer of data between two clients is performed in a point-to-point manner. From the point of view of storing data in a P2P network, it is performed in several places to provide redundancy, which means that a data packet is in several places in the IoT network. Thus, in case of a cyber-attack on any node in the network, the data will be able to be reconstructed through the other copies from the network.

The other feature is decentralization. In classic server-client structures, the server acts as a central data distribution system, while in a P2P structure, this task is managed by the entire network. Thus, the decentralization feature appears, where several nodes have decisions about what is happening in the network.

*C*.Consensus mechanism

A consensus mechanism [28] is a way to gain agreement, trust, and security in a decentralized network based on blockchain technology. The role of consensus mechanisms in a network that uses blockchain as a core component is to ensure that all nodes in the network follow the same rules. If a blockchain has no consensus mechanism, it means that the blockchain has no rules and can become an easy target for malicious entities. Users of a blockchain that works without a set of rules will find it difficult to prove the validity and integrity of the data that are added and shared on the network.

There are currently three consensus mechanisms [29] with the highest percentage of use, being reference mechanisms for the implementation of a blockchain. This is mainly due to the field of application of the technology, namely the financial field. The three consensus mechanisms are Proof of Work (PoW), Proof of Stake (PoS), and Delegated Proof of Stake (DPoS).

Even though blockchain is a technology that has many positive features, it also has some limitations such as large storage capabilities, low scalability, and high energy consumption. When we talk about storage capacity as one of the limitations of blockchain technology, we refer to the fact that the blockchain is constantly growing, i.e., a block of data of a certain size (1 MB in the case of Bitcoin blockchain or 4 MB in the case of Ethereum blockchain) is constantly added to the blockchain, and its size can easily reach the terabyte range. We consider that using state-of-the-art storage solutions, this disadvantage can be overcome.

The low scalability and the energy consumption issues can be solved by the integration of the different consensus mechanism such as PoS that allows support for integrating many IoT devices with low computational recourses.

## 4. Data Storage for IoT Devices Using Blockchain Technologies

Blockchain technology will allow IoT networks to move away from the classic structure of a WSN network, where nodes communicate with a centralized server, used for data storage, authentication, and sensor authorization, to a more secure and flexible approach. Blockchain offers a new method of decentralized and distributed management of both data and nodes in a P2P network. In our system, WSN nodes have full control over the network, which means that any new WSN that wants to use the network must be accepted by the other WSN enrolled in the consensus mechanism. The architecture of the wireless sensor network based on the blockchain technology proposed can be seen in Figure 2. The proposed architecture integrates a novel modular technique that involves using a lightweight blockchain and public blockchain for WSN data storage.

The main entities of the developed architecture are:

**Blockchain Capabilities:** As shown in Figure 2, in the proposed architecture, there are two blockchains, a lightweight blockchain and a public blockchain. The lightweight blockchain is located at the level of the IoT gateways with blockchain capabilities and has the role of storing a light copy of the public blockchain for backup reasons and acts as a register. The public blockchain stores all the data that are sent by the WSN nodes. The lightweight blockchain integrates the node identity ledgers and stores the hash address that points to the data packet itself in the public blockchain. Each time a new block is added to the public blockchain, only the information about the total number of data packets, the validator node ID, and the address of the added block will be saved in the lightweight blockchain. The data block will be saved in the public blockchain.

The public blockchain functions as a database that stores all the data received from the WSN nodes but also stores the information related to the authentication and registration of nodes in the network. In addition, the public blockchain is a P2P system represented by BC storage entities where each one of them has a complete copy of the entire system. The need to have such an entity that has complete copies of the entire blockchain is a method of redundancy in such a system. Usually, these entities have high computing and storage capabilities. Therefore, if a large part of the network nodes become inaccessible and data are lost, the entire system can be rebuilt using a single node that holds a complete copy of the blockchain.

**Blockchain-enabled gateway (BCeGW):** BCeGWs have the role of communicating directly with the WSN to collect the data sent and redirect them further for storage to the public blockchain. Communication between the BCeGW and consensus WSN nodes is performed when a new node wants to join the wireless sensor network and a validator WSN node must be chosen from the list of available ones. The validator WSN node in a concessions WSN node is mains-powered and has a higher computational power than a regular WSN node. At that point, BCeGWs will send the information requested by the consensus node about the new node that wants to join. In addition, the BCeGWs have a local copy of the blockchain, more precisely a lightweight copy of the public blockchain. Another role of BCeGWs is to implement the network consensus mechanism, which is initiated when a new block is added to the public blockchain. As with consensus nodes, a BCeGW will be chosen through the PoS mechanism to validate and add a block to the public blockchain.

**WSN nodes:** At the WSN level, there are 2 types of nodes, sensor nodes and consensus nodes. Each type of node has its role in the network.

**Sensor nodes:** These WSN nodes are registered in the network by the user and have the role of collecting data from the environment. The WSN sensor nodes are devices with low computing power and limited battery level, which transmit information to the BCeGW at user-specified time intervals. The communications of these types of nodes are bidirectional because they will receive specific information from the WSN consensus nodes.

**Consensus nodes:** These WSN nodes are different from sensor nodes in that they also collect data from the environment but are used to implement the consensus mechanisms. These nodes are usually mains-powered. Although the consensus mechanism used in this architecture does not require high computing power, the use of sensor nodes for the implementation of the PoS mechanism would lead to unwanted power consumption. The communications of these nodes are bidirectional, i.e., they can send but also receive information from the BCeGW.

**Consensus model:** The consensus model that is used in the proposed IoT blockchain is the core of the entire P2P network. The consensus mechanism used in a WSN determines the performance of the entire system, which includes throughput, security, and delay. Given that IoT devices do not have high computing power, the implementation of the PoW consensus mechanism is not the most appropriate solution when it comes to IoT devices. To take advantage of many IoT devices from WSNs, in our architecture, we propose the use of a customized PoS consensus mechanism. In the case of the PoS mechanism, the consensus nodes in the WSN must stake their reputation points to increase the chance of being chosen as validators for the data packets that are going to be added in blocks. Once a WSN consensus node has been selected as a WSN validator node, it has the task of validating all the data packets in the block, and then it will have to add the new block to the blockchain. After the consensus node adds the new block to the blockchain, it will be rewarded with reputation points. If the validator WSN node accidentally accepts data packets that contain unreal information, it will lose some of the reputation points, thus making the node less trustworthy. Therefore, the PoS consensus mechanism does not require much computing power, due to its mode of operation, and is a good security solution for a P2P system where nodes are stimulated to make decisions that benefit the entire network and its users. PoS is slowly becoming the most used consensus mechanism among public state-of-the-art blockchains. The PoA mechanism has the below disadvantages [30]:The identity of the validating nodes in the blockchain network is known and this can cause manipulation and interference by third parties for their own benefit.The PoA consensus architecture model has a lower degree of decentralization due to the use of validators nodes. The use of PoA affects the scalability and high throughput of the blockchain architecture.The PoA consensus model is prone to “Sybil” [31] and “Cloning” [32] attacks, where attackers can manipulate a large number of validator nodes by forging multiple identities.

Although the PoA consensus mechanism can be used in public blockchains, its applicability is still largely used in private blockchains that require permissions and test nets such as Kovan [33], Goerli [34], and Rinkeby [35].

**Smart contract mechanism**: In general, a smart contract (SC) [36] is a computer program that is in the blockchain and involves the execution of predefined functions. Once the SC has been implemented in the blockchain, it will receive a unique address (hash value) so that its functions can be called. Smart contracts are also visible to all network participants, but this is not a problem, because they are in compiled bytecode format and cannot be modified.

In our proposed architecture, the SC is implemented at the public blockchain level. The SC functions are built for the WSN ecosystem, for example, WSN sensor nodes registration and the communication between the public blockchain and BCeGW. By using a smart contract, the interaction steps between the BCeGW and the public blockchain are automated and secure. Another important aspect of SC is that the result of the interactions will be a predetermined one, and the possibility of errors is quite small. Because the SC is in the blockchain, there is no way to upgrade or add new features to the source code. If, in the future, there is the need to add new features to the smart contract, this is only possible by changing and relaunching the modified SC in the blockchain. Once the new contract has been launched on the blockchain, all the entities in the system proposed by us will have to use the hash address of the new smart contract to use its new functions.

**Tertiary entities:** In our architecture, there is also the possibility to add standard WSN gateways, which will be known as tertiary entities, without blockchain capabilities that receive information from wireless sensor networks. In the case of these types of WSNs, the standard WSN gateway will also communicate with the smart contract to send data from the sensor nodes for storage. These gateways will not be involved in enforcing consensus mechanisms when new blocks are added to the public blockchain, but the consensus mechanism will be used by the BCeGW when these standard gateways want to add new data packets to the blockchain. Joining new nodes to a classic WSN is performed without the need to use consensus nodes to implement the PoS consensus mechanism. In addition, the type of communication protocols that can be used depends on the standard WSN gateway, ranging from WiFi to NB-IoT. As our proposed system uses both blockchain-enabled gateways (BCeGWs) and traditional gateways without blockchain capabilities, scalability is not an issue. Regardless of the type of gateway but also of the communication protocol, any IoT provider or WSN can join, benefit, and also contribute to the security and smooth functioning of the whole ecosystem easily.

Implementing the blockchain technology with characteristics such as immutability, timestamping, unanimity, distributivity, and decentralization, and with components such as consensus mechanisms, decentralized networks, and the smart contract is a good solution because it can alleviate the security and scalability concerns for IoT.

The blockchain ledger is distributed. This aspect provides some certainty that the data in the blocks will not be altered or deleted, because there is no single entity in control of the network. Blockchain provides transparency. All data transactions made through the blockchain are in plain sight and can be viewed by anyone. This can be a method of identifying a particular source that has added data to the blockchain or even identifying the source where there are data leaks.

The use of blockchain can be another layer of security for the IoT, a layer that third parties need to overcome in order to alter or steal user data.

Thousands, hundreds of thousands, or even millions of IoT packets can be carried out in an automated, secure, and transparent manner without human user intervention. This can be performed through smart contracts.


**WSN authentication scheme**


As previously specified, both consensus nodes and BCeGWs will implement the network consensus mechanism. Consensus nodes implement the PoS mechanism when a new node wants to join the network, and the BCeGW when a new block is added to the public blockchain. The authentication scheme and the steps performed by each entity can be seen in Figure 3.

1. When a new node wants to join a WSN, it will first send a Join Request to the BCeGW and a secure radio communication channel will be established.

2. Once a communication channel has been established, the BCeGW will query the new node to send it a specific data set, such as MAC address, unique ID, and manufacturer ID.

3. The new node will send the entire list of its characteristics requested by the BCeGW.

4. The BCeGW will send the new node’s characteristics to be validated by the WSN consensus node, which will be chosen by applying the PoS consensus mechanism. Also here, one WSN consensus node will be chosen as a validator to verify the information from the node that wants to join the network.

5. After a WSN consensus node has been selected as a validator, it will ask the BCeGW to check if the data for the new node are not already in the local blockchain.

6. The BCeGW will interrogate the local blockchain for the data requested by the WSN consensus node.

7. The local blockchain will return the data if any are available; otherwise, nothing will be returned.

8. The BCeGW will send to the WSN consensus node the result queried from the local blockchain.

9. The WSN consensus node will compare the result from the new node with the result that came from the local blockchain and then it will return a response if the new node is or is not accepted to join the WSN.

10. The BCeGW will send the join request result to be stored on the public blockchain.

**Users:** In the proposed architecture, users can register WSN nodes, BCeGWs, and even BC storage entities that have the role of maintaining a complete copy of the public blockchain. An important thing to note is that the user can only register their BCeGW or full node after using the network for some time and its reputation level is a positive one. To register a BCeGW or a full node, the other users of the network must verify the request, and this is performed automatically through smart contracts that are at the level of the public blockchain. If the application for registration of a BC storage entity or BCeGW is rejected, the user cannot repeat the same application until after a certain period, for example, one week or 6 months. In addition, any registration of a BC storage entity or BCeGW must be paid by the user using reputation points, and if the application is rejected, the points will be lost. When it comes to recording a WSN sensor node, the process to be accepted is one with a higher success rate and no reputation points are needed.

Depending on the number of data packets transmitted by an IoT device using the proposed architecture, the device owner is required to pay a fee for storing the data in the blockchain. This aspect is applied only to the sensor nodes. Another way to pay for the possibility of storing data in the blockchain is by converting reputation points into messages that can be transmitted.

**Reward system:** To motivate users to enroll validator WSN nodes, a rewarding system, in the form of reputation points, can be used in the network. The rewarding system will be implemented at the WSN level, BCeGW level, and full node level. The nodes in the WSN network will receive a certain number of points depending on the type of node and the number of tasks performed in the blockchain architecture. These points can be converted into messages that give the user the possibility to store more data packets in the blockchain.

**API:** The query of the blockchain will be performed through an application programming interface (API), which has the role of returning the data packets requested by users. The only type of permission that the API has in the proposed architecture is to query the blockchain (it only makes GET requests), without the ability to add data (it cannot use POST requests). The necessary information that the API needs to return the data requested by the user is the gateway ID where the sensor is located, and the sensor ID. To return data as soon as possible, it is recommended that the user provides both parameters required by the API.

## 5. Performance Evaluation

For the performance evaluation, we conducted a test setup where we used blockchain technology to securely store data from sensor nodes in WSNs. To perform this test setup, BlockSim [37] was modified and used on a computing system that has an AMD Ryzen 5 1600X processor and 16 GB DDR4 of RAM, over a period of 5 days in total. Thus, as has been specified in Section 4, the proposed architecture does not consider the communication protocol integrated into the IoT network. However, in the case of this paper, to create a test setup as close to reality as possible, we took it into consideration for the emulation of the LoRaWAN communication protocol. According to [38], only 3 gateways are needed to cover an urban area with a radius of about 15 km. A single LoRaWAN gateway is capable of handling up to 100,000 WSN nodes transmitting a data packet of 50 bytes once per hour. The developed architecture considers the analysis of a data storage system whose security is provided by blockchain technology regardless of the communication protocol used. To evaluate the performance level of the proposed architecture, we developed two scenarios that consider both the latency and the throughput of the blockchain architecture. The first metric, latency, is the elapsed time of a data packet that was received by the gateway and when it was added to the blockchain. This performance metric is associated with the processing speed of the proposed architecture. The higher the latency, the more difficult it will be to add data packets into blocks and scale the proposed architecture.

The throughput of the proposed blockchain architecture shows us the obtained performance level when the number of blockchain WSN nodes is increased per BCeGW and represents the total number of processed transactions by the blockchain. As different functionalities of the blockchain are given to BCeGWs, it is important to analyze a load of architecture in different operating conditions to evaluate its scalability. The parameters used for our test setup to evaluate the performance of the proposed architecture included a total number of 5 gateways where the total number of blockchain WSN nodes varied from 500 to approximately 20,000. The size of the block in which the data packets are added was set to a size of 1 MB and the payload had a size of 50 bytes [38].

The performance metric measured in our architecture was the average latency of accepting a data packet in the blockchain, and this included validating and adding it in a block and transaction throughput. As can be seen in Figure 4a, the average latency of accepting a single data packet increases, because the gateway must validate the data packets sent by the blockchain WSN nodes and then add the data packets in blocks. The duration of the validation process of a single data packet increases because the validation process is not performed instantly. Therefore, the data packets will be placed on a waiting list so that they can be validated and then added in blocks, thus increasing the process of validating a single data packet. The validation process for all data packets differs from one data packet to the next. For a more accurate visualization of the performance level, in Figure 4a,b, we can observe the average latency for each performance test.

Following the evaluations, for 500 blockchain WSN nodes, we have an average latency of 55.4 ms, and an average latency of 67.9 ms in the evaluation of 5000 blockchain WNS nodes. In Figure 4b, we can observe that the latency in the test where 7500 blockchain WNS nodes are used is 108.5 ms, and 4261 ms in the test with 20,000 blockchain WSN nodes used.

Figure 5a,b show the transaction throughput, which is the number of transactions that a blockchain architecture can process. In our case, transactions are the data packets sent by the blockchain WSN nodes. In our testing, we have approximately 496.31 TPS for a total of 500 blockchain WSN nodes, and 16,006.73 transactions for a total of 20,000 blockchain WSN nodes.

The final step was to determine the size of the data stored in the blockchain. In this case, parameters such as data packet size, total number of nodes, and frequency of sending data packets were used. The used parameters have characteristics where the data packet size is 50 bytes (mean size of an IoT data packet), the total number of nodes is 20,000, and the frequency of sending data packets is once an hour for each node in the network [38]. If a total of 20,000 nodes each send every hour a 50-byte IoT data packet, in 24 h, we will have a total data volume of 22.8 MB collected from IoT devices. Each IoT packet is stored in the blockchain by means of a transaction.

The data packet transaction contains various information such as sequence ID, input counter, transaction inputs and outputs, output counter, and lock time, which lead to a dimension of about 100 bytes according to [39]. Following this information, the total size of a data packet storage transaction will be about 150 bytes. To summarize, for each 50 bytes of IoT data, we will have 150 bytes stored in blockchain.

Another important aspect that must not be neglected is that the storage capacity of a block in blockchain is not entirely available for storing the IoT data. The block dimensions of 1 MB include information such as block size description, block header, and data transactions counter; thus, approximately 0.95 MB can be used for the data storage transactions [40], [41]. Each block can store up to 6640 packet storage transactions.

From the performed evaluation, we consider the developed storage process to be efficient and scalable due to the PoS consensus integrated mechanism. The proposed architecture can integrate hundreds of thousands of IoT devices distributed over a large geographical area.

## 6. Conclusions

In this paper, we propose an architecture based on blockchain technology that aims to manage and store large amounts of data from IoT devices in wireless sensor networks. By using blockchain technology, the large amounts of data that are sent by the IoT devices will be managed and stored securely, and this is due to the characteristics such as immutability, decentralization, distributivity, and consensus mechanism. The data packets are stored in a distributed manner, thus eliminating the classical centralized storage entity.

The entire architecture is governed by a P2P network, which must ensure proper operation and keep the system functioning. We also consider the fact that IoT devices have limited resources such as memory, computing power, and limited battery capacity, so we propose using the Proof of Stake consensus mechanism that does not require a high level of resources but offers the same security as PoW or DPoS. We also use a smart contract (SC) technique to ensure that the outcome of any information transfer is predefined, thus eliminating the chance of malicious communications.

The main contribution of the proposed architecture is that its scalability is also not being locked on a particular wireless communication protocol. New wireless networks can be easily enrolled in our blockchain architecture without the need for retrofitting. Data storage is performed by using a lightweight blockchain (local blockchain) and a public blockchain. The node joining request to a network is performed using blockchain technologies. We use the Proof of Stake consensus mechanism for the WSN authentication scheme. We propose an architecture where different entities (e.g., gateways) that manage the IoT devices have blockchain capabilities.

In addition, for the architecture proposed in this paper, two characteristics are analyzed, latency and throughput. According to the performance evaluation, the proposed architecture offers low latency, with an average of 55.4 ms for a total of 500 blockchain WSN nodes, and an average of 4.2 s for a total of 20,000 blockchain WSN nodes. In the case of throughput, if we increase the number of blockchain WSN nodes in the network, the architecture scales and can integrate a high number of blockchain WSN nodes. The proposed blockchain architecture uses an IoT authentication process that allows new WSN nodes to be accepted using a voting process that integrates the PoS consensus mechanism. Another advantage is related to the scalability of the proposed architecture, which can integrate a very large number of IoT devices without decreasing the performance level of the system. This IoT devices can join the network and contribute to its maintenance by implementing the consensus mechanism. Our massive blockchain data storage solution can also be used in a hybrid manner by classic IoT WSN networks with no enhanced blockchain capabilities.

Table 1 presents a performance evaluation of the proposed architecture considering other solutions presented in the scientific literature. From the obtained results, the proposed architecture ensures a high level of performance and can be used as a massive data storage solution for IoT devices using blockchain technologies.

As future work, we plan to further test the proposed architecture in real operating conditions in order to evaluate its scalability and efficiency. For better and more rigorous testing, we intend to use application-specific techniques using blockchain technology. Some of these techniques can be found in [41,42].

## Figures and Tables

**Figure 1 sensors-23-01570-f001:**
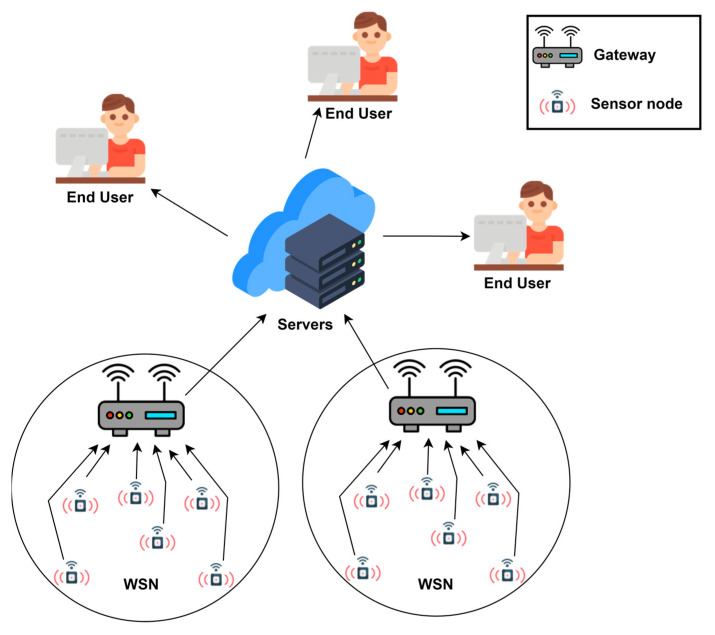
Classic wireless sensor network architecture.

**Figure 2 sensors-23-01570-f002:**
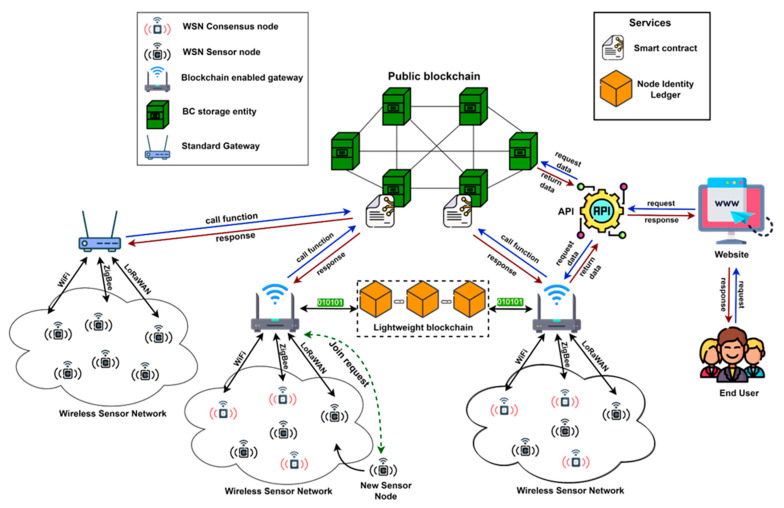
Proposed blockchain WSN architecture for data storage.

**Figure 3 sensors-23-01570-f003:**
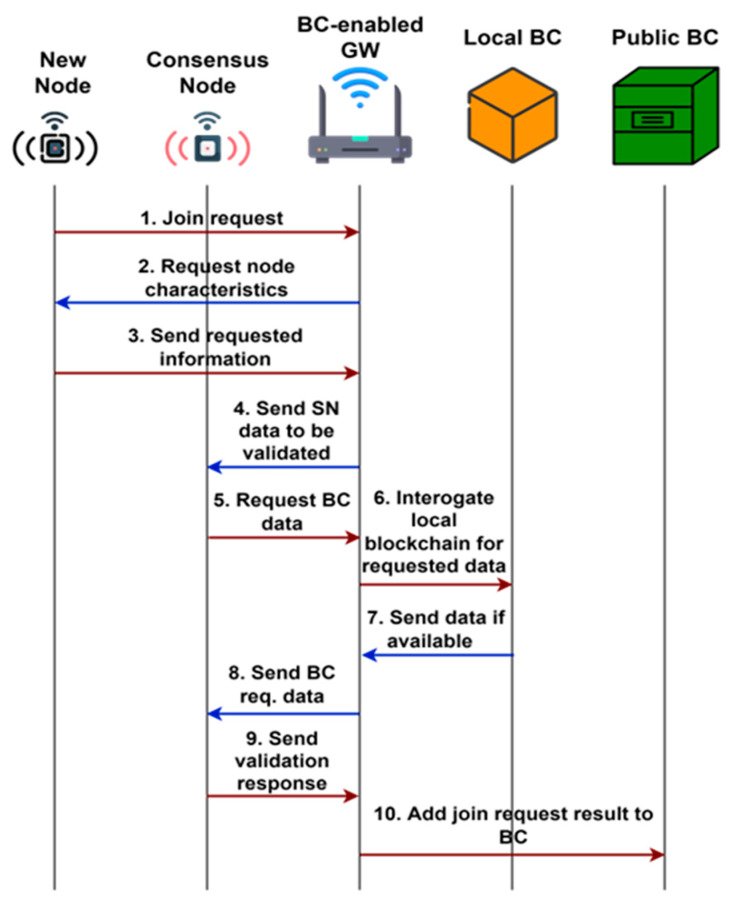
New nodes join the request procedure to a blockchain-enabled WSN.

**Figure 4 sensors-23-01570-f004:**
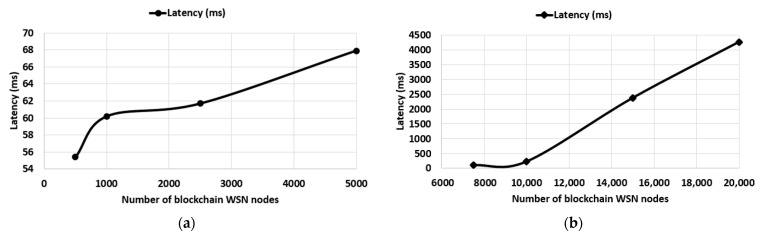
Average latency metric for the proposed blockchain architecture: (**a**) 500, 1000, 2500, 5000 blockchain WSN nodes; (**b**) 7500, 10,000, 15,000, 20,000 blockchain WSN nodes.

**Figure 5 sensors-23-01570-f005:**
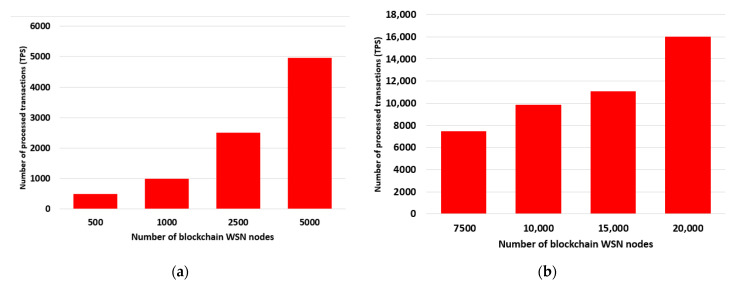
Throughput: (**a**) 500, 1000, 2500, 5000 blockchain WSN nodes; (**b**) 7500, 10,000, 15,000, 20,000 blockchain WSN nodes.

**Table 1 sensors-23-01570-t001:** Performance evaluation of the proposed architecture.

Parameters	Liu et al. [18]	Li et al. [19]	Shafagh et al. [20]	Ren et al. [22]	Ren et al. [23]	Our Approach
Consensus mechanism	PoW	-	PoW	-	PoW	PoS
TPS	12–15	-	12–15	-	12–15	100+
Block Time	-	-	-	-	-	~15 s
Scalable	No	-	No	-	No	Yes
Security	High	High	High	High	High	High
Computational Power	High	-	High	-	High	Low
Storage	High	-	High	-	High	High

## Data Availability

This study did not report any data.

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
