# Peer review of "Massive Data Storage Solution for IoT Devices Using Blockchain Technologies"

_sensors, 2023, doi:10.3390/s23031570_

Round 1

Reviewer 1 Report

Summary: In this study, the authors suggest a blockchain-based architecture for managing and safeguarding data gathered from IoT devices. They do this by integrating blockchain technology, which has benefits including immutability, decentralization, distributivity, greater security, transparency, immediate traceability, and increased efficiency through automation. As a result, there is no longer a need for a centralized network topology. The proposed architecture can be used as a scalable, huge data storage solution for IoT devices employing blockchain technology because the results show that it ensures a high level of performance.

  Comments and Suggestions: - In the introduction, the authors introduce general notions and do not present their contribution covered by this work. After reading the introduction, the reader cannot get any details about the tasks achieved by the authors in this research project.   - The authors may add a table in the related work section summarizing the contributions of previous similar works and providing a comparison with them. The authors need to convince readers about the originality of their work.   - The authors may include some references to survey articles that give an overview of modern applications of blockchain technology, like:
+ https://www.mdpi.com/1424-8220/22/14/5274/xml + https://www.sciencedirect.com/science/article/abs/pii/S2210670720305813 + https://ieeexplore.ieee.org/abstract/document/8656511      - The authors need to provide more justification for the choice of the adopted technologies for implementing the proposed solution.   - It is well known that the Blockchain Technology has many limitations. The authors need to identify these limitations and explain how they intend to tackle them.   - The authors need to report on the cost of the proposed methodology in terms of time and materials.   - As any other technology, Blockchain-Based Systems need to be tested and validated particularly using formal and rigorous techniques. The authors are invited to cover the following references and others for this purpose: + https://www.scitepress.org/Link.aspx?doi=10.5220/0011042800003176 + https://ieeexplore.ieee.org/abstract/document/8813282     - In addition, the authors need to    - In addition, the authors are invited to insert new figures which illustrate the different notions presented in the first sections of the paper.   - The blockchain technology has many limitations. Among these limitations, scalability, the size of data, and the computation time are the major ones. The authors need to explain how they deal with these limitations in an efficient way.   - What types of IoT nodes were used during the experimental work? Does the proposed solution work for types of IoT?    - Was the proposed approach validated in a real environment using real IoT nodes?     - The authors need to provide a strong and reliable estimation of the number of nodes and  the size of data that may be reached by the proposed approach in order to convince the readers about the efficiency of the proposed methodology.

Author Response

Dear Reviewer,

Thank you for your constructive comments concerning the manuscript entitled “The design and development of a microstrip antenna for Internet of Things Applications”. A revision of the paper has been carried out to take all of them into account.

We also appreciate the Reviewer’s time and comments that helped us to improve our manuscript. The revisions are highlighted in the revised manuscript and the point-by-point responses to Reviewer comments are attached.

The comments and suggestions considerably helped us improve the manuscript. We hope our responses answered your questions, and that the new version of the manuscript meets your expectations.

In the process, we believe the paper has been significantly improved. We have studied your comments carefully and made major corrections and extensions.

Please see the attached file for the detailed response.

Lavric Alexandru, PhD

Computers, Electronics and Automation Department

Stefan cel Mare University of Suceava

Reviewer 2 Report

Minor spelling/grammar:

Section 4:

"Blockchain-enabled gateway (BCeGW): BCeGW has the role of communicating directedly with the WSN to collect the data sent..."

Section 5:

Review repeated words (line 398).

Review punctuation in line 416 with regards to "meanwhile,".

Comments:

1. In the simulated evaluation, how are the WSN node and gateway modelled? how would they differ from nodes in the main network where the public blockchain resides? Wouldn't there be a difference in compute capabilities. Also, how is the lightweight blockchain modelled in comparison to the public blockchain? how are they different in the simulation?

2. Of the metric used to evaluate blockchains, why weren't "Transactions per Second", "Block Time" and "number of validators" taken into account?

3. Why was "Proof of Stake" chosen over "Proof of Authority" consensus for validation? As this focuses on application on constrained devices, wouldn't this be useful for comparing, especially with regards to energy efficiency?

4. Among the related work presented, couldn't there be a comparison be made between the proposed approach and that in the related work?

Author Response

(The authors gave the same response as above.)

Round 2

Reviewer 1 Report

The authors considered all my comments and suggestions. I have no more remarks to make. Good luck.